# Perioperative Considerations in Osteogenesis Imperfecta: A 20-Year Experience with the Use of Blood Pressure Cuffs, Arterial Lines, and Tourniquets

**DOI:** 10.3390/children7110214

**Published:** 2020-11-06

**Authors:** Kirsten E. Ross, Joseph T. Gibian, Christy J. Crockett, Jeffrey E. Martus

**Affiliations:** 1Department of Orthopaedic Surgery, Vanderbilt University Medical Center, Nashville, TN 37203, USA; kirsten.ross@vumc.org; 2School of Medicine, Vanderbilt University, Nashville, TN 37203, USA; joseph.gibian@gmail.com; 3Department of Anesthesiology, Vanderbilt University Medical Center, Nashville, TN 37203, USA; christy.crockett@vumc.org

**Keywords:** osteogenesis imperfecta, intraoperative fractures, blood pressure cuff, arterial line, extremity tourniquet

## Abstract

Osteogenesis imperfecta (OI) is a rare genetic connective-tissue disorder with bone fragility. To avoid iatrogenic fractures, healthcare providers have traditionally avoided using non-invasive blood pressure (NIBP) cuffs and extremity tourniquets in the OI population in the perioperative setting. Here, we hypothesize that these procedures do not lead to iatrogenic fractures or other complications in patients with OI. A retrospective study of all children with OI who underwent surgery at a single tertiary care children’s hospital from 1998 to 2018 was performed. Patient positioning and the use of NIBP cuffs, arterial lines, and extremity tourniquets were documented. Fractures and other complications were recorded. Forty-nine patients with a median age of 7.9 years (range: 0.2–17.7) were identified. These patients underwent 273 procedures, of which 229 were orthopaedic operations. A total of 246 (90.1%) procedures included the use of an NIBP cuff, 61 (22.3%) an extremity tourniquet, and 40 (14.7%) an arterial line. Pediatric patients with OI did not experience any iatrogenic fractures related to hemodynamic monitoring or extremity tourniquet use during the 20-year period of this study. Given the benefits of continuous intra-operative hemodynamic monitoring and extremity tourniquets, we recommend that NIBP cuffs, arterial lines, and tourniquets be selectively considered for use in children with OI.

## 1. Introduction

Osteogenesis imperfecta (OI) is a collection of rare genetic connective tissue disorders that affects bone, tendons, skin, teeth, fibrocartilage, cornea, and endomysium and ultimately causes bone fragility and deformities [1,2,3]. Subtypes of OI are based on the specific type of collagen affected and display differing frequency and severity of symptoms, ranging from frequent fractures to intrauterine demise [3,4]. Specifically, type III is the most severe non-lethal form of the more common autosomal dominant types [3]. There are also less common autosomal recessive types that have severe phenotypes, such as types VII–X and XII [3]. These patients typically have a history of multiple fractures from infancy, significant bone deformities, and short stature [3]. Other symptoms of OI include hearing loss, blue sclera, kyphosis and scoliosis, vascular anomalies, poor dentition, and cardiovascular compromise [2,3].

OI patients typically require many operative interventions throughout childhood for their fractures and bony deformities. It is common for patients to undergo general anesthesia for these procedures, which requires intraoperative blood pressure monitoring [5,6,7]. Anesthetic management of OI patients is influenced by the co-existing orthopaedic deformities and the potential for additional fractures during the perioperative time period [8]. Some (not all) anesthetic considerations in OI patients are included in the following information. Succinylcholine administration is avoided due to the possibility of muscle fasciculations causing bone fractures [8]. The fragility of bones and connective tissue demands extreme care in padding and positioning during anesthesia [9]. It is believed that inflation of automated blood pressure cuffs may be hazardous and can result in bone fractures [8]. However, arterial blood pressure (BP) is a fundamental cardiovascular vital sign and a critical part of monitoring an anesthetized patient; standards for basic anesthetic monitoring mandate measurement of arterial blood pressure at least every 5 min in all anesthetized patients (with the exception of extenuating circumstances, which would require documentation of reasoning in the medical record) [9,10]. There is anesthesia literature that has cautioned against the use of non-invasive blood pressure (NIBP) cuffs in OI patients [11,12]. Insertion of an arterial catheter for longer operations avoids repeated inflation of the cuff and the risk of a bone fracture [9]. In addition, to reduce fracture risk from NIBP cuffs if used, some anesthesiologists will place cuffs on multiple different extremities and check the BP in a rotating fashion every 5 min in order to increase the time between intervals of inflation on a single extremity. Others will only check BP with cuffs intermittently depending on the operation, patient co-morbidities, and current clinical situation.

Tourniquets are also frequently avoided in this patient population due to the iatrogenic fracture concern, which potentially increases blood loss and the need for transfusion. This is common practice despite a recent study showing the relative safety of NIBP monitoring and extremity tourniquets in this patient population [13]. To our knowledge, there is limited evidence that correlates blood pressure cuffs and tourniquets with an increased risk of intraoperative fracture; only anecdotal experience exists.

In this study, our primary aim was to determine the incidence of iatrogenic fractures in OI patients after an operating room encounter over a 20-year period. We hypothesized that there is a low rate of complications associated with patient positioning and the use of blood pressure cuffs, tourniquets, and arterial lines in the perioperative setting.

## 2. Materials and Methods

An institutional review board (IRB) approved retrospective review of pediatric OI patients who underwent an operative procedure at a single tertiary care center was performed. Patients were identified from billing records by the diagnostic codes for osteogenesis imperfecta, ICD-9 756.51 or ICD-10 Q78.0. Inclusion criteria was as follows: documented diagnosis of osteogenesis imperfecta, age less than 18 years at the time of surgery, and surgery between 1 January 1998 and 1 September 2018. Patients were excluded if there were inadequate postoperative records. A waiver of consent was granted by the IRB for this retrospective chart review since no Health Insurance Portability and Accountability Act (HIPAA) identifiers were collected. Data was collected from the medical record (electronic and paper charts) and stored on password-protected, encrypted spreadsheets. Demographic data collected included patients’ age at the time of operation, sex, and OI subtype. Procedure data collected included procedure type, proceduralist specialty, NIBP use and location, arterial line use and location, patient positioning, and orthopaedic extremity tourniquet use, location, pressure, and duration. Complications were identified by searching the operative report and postoperative inpatient notes, outpatient notes, and radiology reports. All statistics (mean, median, range) were calculated using Microsoft Excel.

## 3. Results

### 3.1. Patient Characteristics

Forty-nine patients with a median age of 7.9 years (range: 0.2–17.7) were included in this study (Table 1). A total of 51.7% of patients were female. Seventeen of the 49 patients were formally diagnosed with OI type I, which was the most common subtype in the cohort. Fifteen patients were type III, 12 were type IV, and 1 had OI type VIII. The specific subtype of OI was not documented in four patients. Combined, the 49 patients underwent a total of 273 procedures (mean 5.6 procedures/patient). Two-hundred and twenty-nine of these procedures were performed by the orthopaedic service, while 44 were performed by non-orthopaedic services. Common non-orthopaedic services that performed procedures in the cohort included general surgery, dentistry and oral surgery, urology, neurosurgery, and obstetrics and gynecology.

### 3.2. Intra-Operative Hemodynamic Monitoring

Of the 273 procedures included in our study, 246 (90.1%) involved the use of an NIBP cuff to monitor blood pressure intraoperatively (Table 1). The most common site for NIBP use was the upper arm, followed by the leg and the thigh. The specific site of use was not recorded in 64 of the procedures. An arterial line was used in 40 (14.7%) procedures. Sixteen (5.9%) procedures involved using both an NIBP and an arterial line. Arterial lines were most frequently placed in the radial artery (85%), followed by the ulnar artery (7.5%), posterior tibial artery (5.0%), and femoral artery (2.5%). In eleven (4.0%) procedures, neither an NIBP nor an arterial line was utilized.

### 3.3. Procedure Characteristics

Two-hundred and thirty-eight (87.2%) of the 273 procedures were performed with the patient in the supine position, accounting for the majority of procedures in the study. In order of descending frequency, patients were also positioned in the prone, lateral decubitus, lithotomy, sloppy lateral, and semirecumbant positions. In addition, five patients were re-positioned intra-operatively; three were positioned both supine and lateral decubitus, one in the supine and lithotomy positions, and one in the supine and sloppy lateral positions.

A tourniquet was utilized in 61 (22.3%) of all procedures (Table 1). A total of 80 tourniquets were used in these 61 procedures, related to bilateral procedures. Of these 80 tourniquets, 65 (81.3%) were on the thigh, 113 (16.3%) were on the upper arm, and 2 (2.5%) were on the lower leg. The median tourniquet time was 69 min (range: 10–120 min), with a median tourniquet pressure of 200 mm Hg (range: 180–300 mm Hg). A total of 55% of patients never had a tourniquet used intra-operatively; there was no trend regarding tourniquet use or non-use based on subtype of OI.

### 3.4. Complications

There were no iatrogenic fractures identified secondary to NIBP use, extremity tourniquet use, or patient positioning in our cohort. There was one iatrogenic fracture that occurred intra-operatively during the removal of a flexible intramedullary nail in the affected bone; this fracture was a result of the procedure and a tourniquet had not been utilized. The fracture was recognized intra-operatively and was treated in a long arm cast for 6 weeks with no long-term sequelae related to the injury. No complications related to arterial line usage or intraoperative positioning were identified.

## 4. Discussion

When treating patients with OI, anesthesiologists and surgeons have traditionally been conservative with the use of NIBP cuffs and extremity tourniquets. At our institution, we have historically been less hesitant to utilize NIBP cuffs and extremity tourniquets on these patients (90% of these procedures utilized NIBP cuffs). Anecdotally, we did not experience complications when these were used in the perioperative setting, and they have been used at the discretion of the attending surgeon and anesthesiologist. After collecting data on our cohort of OI patients, we did not identify any iatrogenic fractures related to the use of these devices. This suggests that the risk of fracture is relatively low. Of note, our cohort included 15 patients undergoing 90 procedures with type III OI. Of these procedures, 11.1% utilized an extremity tourniquet and 74.4% utilized NIBP cuff (Table 1). Our data shows that even with severe phenotypes, NIBP cuffs and extremity tourniquets may be safely used with a low risk of iatrogenic fracture.

Hemodynamic monitoring and minimization of bleeding are important for patients undergoing an orthopaedic or non-orthopaedic procedure. Blood pressure monitoring can be achieved through non-invasive or more invasive techniques, such as arterial lines. The latter has been shown to be safe and with a low rate of complication; however, it adds time and cost to an operating room encounter [14,15]. Both the anesthesiologist and surgeon benefit from accurate monitoring of the patient’s blood pressure, whether that information comes from non-invasive or invasive monitoring. Monitoring of patients under anesthesia is critical to provide an early warning of adverse trends or changes hemodynamically before irreversible damage occurs [16]. For the anesthesiologist, this monitoring ensures that if the blood pressure approaches critical levels, the proper interventions may be initiated, such as fluid resuscitation, vasopressors, or anti-hypertensives [17]. For the surgeon, control of bleeding in the operative field is critical when performing any operation. Excessive hypertension can contribute to increased blood loss. Poorly controlled bleeding may lengthen the procedure and place the patient at increased risk for transfusion. Extremity tourniquets can reduce blood loss and expedite an operation.

Similar to our study, a recent single-center study of 37 OI patients aged 21 years and younger who underwent 98 procedures documented no iatrogenic fractures related to intra-operative tourniquet or NIBP monitoring [13]. There were no complications related to arterial lines, while there was one intraoperative fracture associated with patient positioning. Although our study is similar in methods, our patient cohort is larger with significantly more procedure encounters identified. Our study adds to this growing body of evidence that blood pressure cuffs and extremity tourniquets can be utilized safely in the OI population. To our knowledge, there are no published reports intra-operative iatrogenic fractures secondary to the use of NIBP cuffs or tourniquets in patients with OI.

Based on this collective experience, we recommend that pediatric OI patients be selectively considered for intraoperative hemodynamic monitoring with NIBP cuffs and extremity tourniquet usage.

Limitations of this study include its retrospective nature, which lends itself to potential errors and omissions in documentation. All data used in this study was transcribed from standardized operative reports and anesthesia care records, which decreases the likelihood of errors in documentation when compared to general documentation. There is no protocol at our institution to obtain radiographs on extremities that have been used for NIBP cuffs or tourniquets, so there is the potential that a fracture could have been missed. However, OI patients and families will typically notify their orthopaedist when they experience substantial new pain that may represent a fracture. In addition, the specific precautions to lessen the risk of fracture related to an NIBP cuff were variable dependent upon the individual anesthesiologist. For example, some anesthesiologists at our institution will place padding under NIBP cuffs or place cuffs on multiple different extremities and check the BP in a rotating fashion every 5–15 min in order to increase the time between intervals of inflation on a single extremity. Some will only use NIBP cuffs while placing an arterial line and then utilize the arterial line only for the remainder of the case. Others will only check NIBP intermittently depending on the operation, patient co-morbidities, and current clinical situation. These details are not always included in the anesthetic record and are a limitation to our study. It is possible that these variations in decreased frequency of measuring NIBP on a single extremity could decrease risk of fractures, but our review still suggests that risk of fracture is relatively low with NIBP measurements. These precautions may be considered as useful techniques if others wish to increase utilization of NIBP measurements in this population.

In addition, as OI is a rare disease, the sample size of this study is relatively small with 49 patients having undergone 273 procedures, which may lead to a type II error when considering iatrogenic fractures. However, this study is the largest series to-date that investigates iatrogenic fractures associated with NIBP and tourniquet use in patients with OI. Finally, this study was performed at a single institution; however, the techniques for blood pressure monitoring and the tourniquet application likely do not vary substantially between geographic locations.

## 5. Conclusions

Traditionally, the use of NIBP cuffs and extremity tourniquets has been limited over fears of iatrogenic fractures in OI patients. Here, we demonstrate no intra-operative iatrogenic fractures or other complications in this population over a 20-year period, indicating that patients with OI may be selectively considered for the standard hemodynamic monitoring and extremity tourniquet usage.

## Figures and Tables

**Table 1 children-07-00214-t001:** Demographics and Procedural Characteristics.

OI Type	Patients	Mean Age (years)	Procedures	Orthopaedic Procedures	Non-Orthopaedic Procedures	% Procedures with Tourniquet	% Procedures with NIBP Cuff	% Procedures with Arterial Line
I	17	8.8	88	81	7	31.8%	100%	5.7%
III	15	7.9	90	69	21	11.1%	74.4%	30.0%
IV	12	7.8	85	76	9	25.9%	100%	8.2%
VIII	1	1.6	4	1	3	0%	0%	0.0%
Unknown	4	9.7	6	2	4	16.7%	100%	16.7%
Total	49	8.1	273	229	44	22.3%	90.1%	14.7%

OI: Osteogenesis imperfecta; NIBP: non-invasive blood pressure.

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
