# Peer review of "Perioperative Considerations in Osteogenesis Imperfecta: A 20-Year Experience with the Use of Blood Pressure Cuffs, Arterial Lines, and Tourniquets"

_children, 2020, doi:10.3390/children7110214_

Round 1

Reviewer 1 Report

The authors review children with osteogenesis imperfecta who underwent surgeries over a 20 year period to look for complications related to blood pressure cuffs, arterial lines, and tourniquets. The manuscript is well-written and suggests that the use of these devices is generally safe within this population given that no iatrogenic fractures were identified.

A few questions and points to clarify:

It is interesting to me that the authors state that "healthcare providers have traditionally avoided NIBP cuffs and extremity tourniquets in the OI population," and yet NIBG cuffs were used in >90% of the procedures studied. Has the field already moved away from this traditional avoidance, or are the authors at an institution that has been more aggressive in their use, or something else?

The authors cite another recent publication with the same findings - it may be useful to include some additional discussion about how the studies compare and how this study adds additional value to the literature.

Was there any standardized post-operative evaluation to look for fractures in the affected extremity (where the NIBP cuff or tourniquet had been placed)? Like routine x-rays? If not, do the authors believe there is potential for missed fractures?

Did all the patients WITHOUT use of NIBP cuff have an arterial line placed?

The authors note that the "specific precautions to lessen the risk of fracture related to a NIBP cuff were variable" depending on the anesthesiologist. The audience may benefit from knowing some of the specific precautions that were used. Based upon their review, do the authors have any recommendations for precautions to take if others wish to use NIBP cuffs more frequently in this population?

Reviewer 2 Report

This small, single centre retrospective audit of 20 years experience of OI patients investigates the documented fractures in these patients. The authors did not detect any iatrogenic problems in this cohort.

The manuscript is easy to follow and of appropriate length. The introduction can be shortened by a third.

There is a discrepancy in the patient numbers reported in the abstract and the results (61 versus 49 patients).
